# *Neopestalotiopsis siciliana* sp. nov. and *N. rosae* Causing Stem Lesion and Dieback on Avocado Plants in Italy

**DOI:** 10.3390/jof8060562

**Published:** 2022-05-25

**Authors:** Alberto Fiorenza, Giorgio Gusella, Dalia Aiello, Giancarlo Polizzi, Hermann Voglmayr

**Affiliations:** 1Department of Agriculture, Food and Environment (Di3A), University of Catania, Via S. Sofia 100, 95123 Catania, Italy; alberto.fiorenza@phd.unict.it (A.F.); giorgio.gusella93@gmail.com (G.G.); gpolizzi@unict.it (G.P.); 2Department of Botany and Biodiversity Research, University of Vienna, Rennweg 14, 1030 Vienna, Austria; hermann.voglmayr@univie.ac.at

**Keywords:** *Neopestalotiopsis*, *Persea americana*, fungal diseases, stem lesion, pestalotioid fungi, phylogeny

## Abstract

Avocado (*Persea americana*) represents an important emerging tropical crop in Italy, especially in the southern regions. In this study, young plants of avocado showing symptoms of stem and wood lesion, and dieback, were investigated. Isolations from symptomatic tissues consistently yielded colonies of *Neopestalotiopsis*-like species. The characterization of representative isolates was based on the observation of morphological characters, the effect of temperature on mycelial growth rate, and on the sequencing of three different gene regions, specifically ITS, *TEF1*, and *TUB2*. Phylogenetic analyses were conducted based on maximum parsimony and maximum likelihood approaches. The results showed the presence of two species, viz. *Neopestalotiopsis rosae* and *N. siciliana*, the latter of which is here described as a new species. Pathogenicity tests were conducted using the mycelial plug technique on young potted avocado trees for both *Neopestalotiopsis* species. The results showed that both species were pathogenic to avocado. This study represents the first report of these two species affecting avocado and results in the description of a new species within the genus *Neopestalotiopsis*. Based on phylogeny, *Pestalotiopsis coffeae-arabicae* is combined in *Neopestalotiopsis*.

## 1. Introduction

Avocado (*Persea americana* Mill.) is a tree native to Central America and is widely cultivated, especially in tropical and subtropical regions. In the recent years, the consumption and new plantings of this tropical fruit are increasing world-wide. Global top producers (1000 metric tons unit, year 2020) include Mexico (2390), Colombia (876), and the Dominican Republic (676) [1]. Of European countries, Spain is the main avocado producer (99), followed by Greece (10) [1]. In southern Italy, specifically in Sicily, avocado represents an emerging crop in terms of economic opportunity for the growers. In recent years, an increasing consumer interest towards tropical fruits has been observed. Within this new trend, avocado fruit presents great potential due to its high nutritional value and peculiar quality characteristics to achieve requirements desired by consumers [2]. In Italy, since this crop is emerging, few studies have been conducted on the phytopathological situation. Regarding the fungal diseases affecting avocado, several fungal taxa have been reported associated with different symptoms [3], and some of those pose a severe threat for its production around the world. Among these, *Phytophthora cinnamomi* is considered the most important and widely spread pathogen of avocado, causing significant economic losses [4]. *Rosellinia necatrix*, the causal agent of white root rot, is a serious threat in the Mediterranean area and is considered the main cause of avocado losses in Spain [5], and recently it has also been reported in Italy [6]. In Italy, as well as around the world [7], several species belonging to the Nectriaceae family (i.e., *Cylindrocladiella peruviana*, *Ilyonectria macrodidyma*, and *Pleiocarpon algeriense*) have been studied in the last few years, and have been associated with root and crown rot [8,9]. Several fungal species belonging to Botryosphaeriaceae and Diaporthaceae families are known to be causal agents of branch canker and fruit stem-end rot on avocado [10,11,12,13]. In addition, in 2018, a new species named *Neocosmospora perseae* was described as causing trunk cankers in Italy and more recently in Greece [14,15]. *Colletotrichum* spp. are reported as important pre- and post-harvest pathogens [16,17,18,19]. Moreover, pestalotioid fungi, known as major agents of leaf spot diseases, were also reported on avocado [20,21,22]. During December of 2020, surveys conducted in a commercial avocado orchard in Sicily (Italy) revealed the presence of young plants showing external symptoms of dieback and stem lesions on scion at the grafting point or slightly above. Since avocado is considered an emerging crop in Italy, especially in the southern regions, it is crucial to investigate the etiology of the diseases that could represent an important limiting factor. The aim of the present study was to investigate the etiology of the symptoms observed in the field and to identify the causal fungal agents to species by morphology and molecular data.

## 2. Materials and Methods

### 2.1. Isolation and Morphological Characterization

Samples were collected in the field on approximately 20 plants of *Persea americana* cv. “Hass” grafted on “Zutano”, randomly selected, and were brought to the Plant Pathology laboratory of the Department of Agriculture, Food, and Environment at the University of Catania for further investigations. One hundred small sections (5 mm × 5 mm) of the stems were surface disinfected for 1 min in 1.5% sodium hypochlorite (NaOCl), rinsed in sterile distilled water, dried on sterile absorbent paper, and placed on potato dextrose agar (PDA; Lickson, Vicari, Italy) amended with 100 mg/liter of streptomycin sulphate (Sigma-Aldrich, St. Louis, MO, USA) to prevent bacterial growth, and then incubated at 25 ± 1 °C for five to seven days. Single-spore isolates were obtained from conidia produced in pure cultures grown on PDA. To determine the effect of temperature on mycelial growth and the optimal growth temperature, the representative isolates AC46 and AC50 were cultured on PDA for further assays. After seven days of incubation at 25 °C, 5 mm diam. mycelial plugs were transferred from the edge of the colonies to the center of PDA Petri plates. The plates were incubated at 5–10–15–20–25–30–35 °C in the dark. Three Petri plates were used for each temperature as replicates. The experiment was repeated once. After seven days of incubation, two perpendicular diameters of the colonies were measured with a scale ruler. The isolates used in this study are maintained in the culture collection of the Department of Agriculture, Food, and Environment, University of Catania. Moreover, representative isolates were deposited at the Westerdijk Fungal Biodiversity Institute (CBS), Utrecht, the Netherlands, and dried sporulating cultures were deposited as vouchers in the fungarium of the Department of Botany and Biodiversity Research, University of Vienna (WU-MYC).

Study of macromorphology of conidiomata was performed by using a Nikon SMZ 1500 stereomicroscope equipped with a Nikon DS-U2 digital camera or with a Keyence VHX-6000 digital microscope (Mechelen, Belgium). Microscopic preparations were mounted in water. For light microscopy, a Zeiss Axio Imager.A1 compound microscope (Oberkochen, Germany), equipped with Nomarski differential interference contrast (DIC) optics and a Zeiss Axiocam 506 color digital camera, was used. Photographs and measurements were taken by using the NIS-Elements D v. 3.0 or Zeiss ZEN Blue Edition software. For certain images of conidia, the stacking software Zerene Stacker version 1.04 (Zerene Systems LLC, Richland, WA, USA) was used. Measurements are reported as maxima and minima in parentheses and the mean plus and minus the standard deviation of a number of measurements given in parentheses.

### 2.2. DNA Extraction, PCR, and Phylogenetic Analysis

The genomic DNA was extracted from surface mycelium scraped off from pure cultures, using the Wizard Genomic DNA Purification Kit (Promega Corporation, Madison, WI, USA). The following three loci were amplified and sequenced: the complete internal transcribed spacer region (ITS1-5.8S-ITS2) with primers ITS5 and ITS4 [23]; a ca. 0.5 kb fragment of the translation elongation factor 1-alpha (*TEF1*) gene with primers EF1-728F [24] and TEFD_iR [25]; and a ca. 0.95 kb fragment of the beta-tubulin (*TUB2*) gene with primer pairs T1 [26] and BtHV2r [27]. The PCR products were purified using an enzymatic PCR cleanup [28], as described by Voglmayr and Jaklitsch [29], and sequenced in both directions by Macrogen Inc. (Seoul, South Korea) or at the Department of Botany and Biodiversity Research, University of Vienna, using the ABI PRISM Big Dye Terminator Cycle Sequencing Ready Reaction Kit v. 3.1 (Applied Biosystems, Warrington, UK) and the original PCR primers; sequencing was performed on an automated DNA sequencer (3730xl Genetic Analyser, Applied Biosystems). The DNA sequences generated were assembled with Lasergene SeqMan Pro (DNASTAR, Madison, WI, USA). Sequences generated during the present study were deposited in Genbank (Table 1). The newly generated sequences were aligned to a representative matrix of *Neopestalotiopsis*, selecting two species of *Pestalotiopsis* as an outgroup. For *Neopestalotiopsis*, all 70 accepted species were included in the matrix, preferentially with ex-type sequences. The GenBank accession numbers of sequences used in these analyses are given in Table 1.

Sequence alignments for phylogenetic analyses were produced with the server version of MAFFT (http://mafft.cbrc.jp/alignment/server/, accessed on 22 March 2022), and checked and refined using BioEdit v. 7.2.6 [57]. The ITS rDNA, *TEF1*, and *TUB2* matrices were combined for subsequent phylogenetic analyses, and the combined data matrix contained 2265 characters (545 nucleotides of ITS, 900 nucleotides of *TEF1*, and 820 nucleotides of *TUB2*). Maximum likelihood (ML) analyses were performed with RAxML [58], as implemented in raxmlGUI 1.3 [59], using the ML + rapid bootstrap setting and the GTRGAMMA substitution model with 1000 bootstrap replicates. The matrix was partitioned for the different gene regions. For evaluation and discussion of bootstrap support, values below 70% were considered low, between 70 and 90% medium/moderate, and above 90% high and 100% maximum. Maximum parsimony (MP) bootstrap analyses were performed with PAUP v. 4.0a169 [60], with 1000 bootstrap replicates using five rounds of heuristic search replicates with random addition of sequences and subsequent TBR branch swapping (MULTREES option in effect, steepest descent option not in effect, COLLAPSE command set to MINBRLEN, and each replicate limited to 1 million rearrangements) during each bootstrap replicate. All molecular characters were unordered and given equal weight; analyses were performed with gaps treated as missing data; and the COLLAPSE command was set to minbrlen.

### 2.3. Pathogenicity Test

Pathogenicity tests were carried out by artificial inoculations using the isolates AC50 (*N. rosae*) and AC46 (*N. siciliana*). Three potted 1-year-old plants of avocado cv. “Hass” grafted on “Zutano” were inoculated with each isolate. Inoculations were made on the stem after removing a piece of bark with a sterile scalpel blade, placing a mycelial plug (0.3 cm^2^) from a 15-day-old culture of each isolate onto the wound and covering it with Parafilm^®^ (American National Can, Chicago, IL, USA) to prevent desiccation. The same number of plants was inoculated with sterile PDA plugs to serve as control. All the inoculated plants were grown in a growth chamber with a 12 h photoperiod and maintained at 25 ± 1 °C. The inoculated plants were monitored weekly for development of symptoms, and a final assessment was conducted 50 days after the inoculations. Re-isolations were performed as described above to fulfill Koch’s postulates.

### 2.4. Data Analysis

Data derived from the effect of temperature on mycelial growth rate assay and the lesion length measurements were analyzed in Statistix 10 (Analytical Software 2013) [61]. For analysis of the effect of temperature on mycelial growth, variances of the two assays were tested for the homogeneity using Levene’s test and then combined in one dataset. Data of the mycelial growth were first transformed to radial growth rate (cm day^−1^) and then a nonlinear regression adjustment of the dataset was applied through the generalized Analytis β model, using the equation described by Moral et al. [62]. Optimum growth temperature was also calculated according to the equation provided by the same authors [62]. For pathogenicity test, analysis of variance (ANOVA) of the lesions length was performed and the mean differences were compared with Fisher’s protected least significance difference (LSD) test at α = 0.05.

## 3. Results

### 3.1. Isolation and Morphological Characterization

The disease was observed in a commercial avocado orchard located in Giarre (Catania province) on young plants (two years old, 2–3 months after transplanting). Symptoms observed in the field included stem lesions, wood discoloration with brownish streaking, bark cracking, and dieback. Necrotic lesions were characterized by a shrinkage of the affected tissues and internally the wood appeared darkened and dry (Figure 1). Internal lesions started more frequently from the grafting point. The rootstock showed no symptoms. Isolations frequently yielded *Neopestalotiopsis*-like fungi. A total of eight single-spore isolates were collected and kept in our fungal collection. The highest growth rate for the isolate AC46 (1.13 cm day^−1^) was observed at 25 °C. According to the Analytis β model, the optimal growth temperature resulted at 24.6 °C. After 7 days of incubation, no mycelial growth was observed at 35 °C. Isolate AC50 showed the highest growth rate (1.06 cm day^−1^) at 25 °C, and the optimal growth temperature resulted at 21.9 °C. All results of the effects of temperature on mycelial growth rate are shown in Figure 2 and Figure 3.

### 3.2. Phylogenetic Analysis

PCR amplification of the ITS, *TEF1*, and *TUB2* generated 549, 549–550, and 957 bp fragments, respectively. Of the 2261 characters included in the phylogenetic analyses, 334 were parsimony informative (58 from the ITS, 153 from *TEF1*, and 123 from *TUB2*). The best ML tree (−lnL = 8699.596) revealed by RAxML is shown as a phylogram in Figure 4. While backbone support of deeper nodes was mostly absent, several terminal nodes received medium to high support. Of the four *Neopestalotiopsis* isolates of the current study, one was placed within the *N*. *rosae* clade, while the other three isolates were contained within a moderately supported clade together with four unnamed Japanese isolates from *Eriobotrya japonica*. The latter clade was further subdivided into two subclades: a highly supported subclade containing the three isolates of the present study and one isolate from *Eriobotrya japonica*, and a second, moderately supported subclade containing the residual isolates from *Eriobotrya japonica*.

### 3.3. Pathogenicity Test

The results of the pathogenicity test showed that both species of *Neopestalotiopsis* identified in this study were pathogenic to avocado and produced the same symptoms similar to those observed in the field. All inoculated trees showed severe external and internal wood discoloration. Controls did not show any symptoms (Figure 5). For both species, the presence of acervuli on the inoculated wounds was observed. The mean lesion lengths of *N. rosae* (7.76 cm) and *N. siciliana* (6.65 cm) were significantly different from the control (0.6 cm) (*p* < 0.05), but not significantly different between them (Figure 6). Re-isolations showed the presence of colonies with the same morphological characteristics as the inoculated species, so Koch’s postulates were fulfilled. 

### 3.4. Morphological Description of Neopestalotiopsis rosae Isolates from Avocado 

***Neopestalotiopsis rosae* Maharachch., K.D. Hyde and Crous**, in Maharachchikumbura, Hyde, Groenewald, Xu and Crous, *Stud. Mycol*. 79: 147 (2014) (Figure 7).

**Description**—sexual morph unknown. Asexual morph: conidiomata on natural substrate acervular, in culture on PDA sporodochial; solitary, pulvinate, black, 30–100(–150) μm diam., and exuding black conidial masses. Conidiophores indistinct and usually reduced to conidiogenous cells. Conidiogenous cells 1–33 × 1–3.7 μm, discrete, either short-cylindrical, sitting laterally on hyphae, or cylindrical, ampulliform to lageniform, hyaline, smooth, thin-walled, simple, holoblastic-annelidic, proliferating one to two times percurrently, with collarette present and not flared. Conidia (20–)22–24(–25) × (6.2–)6–8.7(–15.2) μm, l/*w* = (1.6–)2.9–3.6(–3.9) (*n* = 40), fusoid, straight or slightly curved, four-septate, smooth, slightly constricted at the septa; the basal cell obconic with a truncate base, thin-walled, hyaline or pale brown, and (3.3–)3.8–4.8(–5.4) μm long; three median cells trapezoid or subcylindrical, (8–)14–17(–17) μm long, smooth-walled, versicolored, with septa darker than the rest of the cell; the second cell from the base pale brown and (3.8–)4.6–5.5(–6.1) μm long; the third cell s medium brown and (4–)4.5–5.1(–5.7) μm long; the fourth cell medium brown and (4.4–)5–5.6(–6.1) μm long, septum between the third and fourth cell more distinct, broader, and darker brown than the other septa; the apical cell conic with the subacute apex thin-walled, smooth, hyaline, (2.8–)3.4–4.4(–4.8) μm long, with two to four apical appendages (mostly three) arising from the apical crest; apical appendages unbranched, tubular, centric, and straight or slightly bent, (15–)19–28(–33) μm long, and (0.8–)0.9–1.1(–1.3) μm wide (*n* = 60); basal appendage single, filiform, unbranched, centric or eccentric, (3–)3.5–5.8(–8.1) μm long and 0.5–0.9 μm wide (*n* = 85). 

**Culture characteristics**. The colony on PDA attaining 90 mm diameter after 7 days at 21.9 °C, yellowish, with a fluffy whitish aerial mycelium, secreting a yellowish pigment in the culture medium, with isolated conidiomata scattered on the aerial mycelium (Figure 8A). The reverse is pale yellowish brown (Figure 8B).

**Habitat**. On stems of *Persea americana* Mill.

**Distribution**. Sicily, Italy.

**Specimens examined**. ITALY, Sicily, Catania Province, Giarre, 15 December 2020, Alberto Fiorenza (WU-MYC 0045984; culture AC50 = CBS 149120).

**Notes**. Our strain AC50 has identical ITS and *TEF1* and highly similar *TUB2* (99.8%; 1 nt difference) sequences to the type strain of *N. rosae* (CBS 101057). *Neopestalotiopsis rosae* has been recorded as a pathogen of various fruit crops, e.g., blueberry (*Vaccinium corymbosum*; [53,63]), pomegranate (*Punica granatum*; [64]), and in particular strawberry (*Fragaria* × *ananassa*), on which it was reported to cause severe disease outbreaks around the world in recent years (e.g., Australia [65], China [66], Mexico [52], Taiwan [67], and the USA [65]). In the protologue of *N. rosae*, it was characterized by three to five tubular apical appendages not arising from the apical crest but at different regions in the upper half of the apical cell. This does not agree with our observations, as in our strain, the apical appendages arise from the apical crest. However, the descriptions and illustrations of the other reports of *N. rosae* cited above agree well with our strain, as do the molecular data. 

## 4. Taxonomy

***Neopestalotiopsis coffeae-arabicae* (Yu Song, K. Geng, K.D. Hyde and Yong Wang bis) Voglmayr, comb. nov**. MycoBank No.: MB 844083.

Basionym: *Pestalotiopsis coffeae-arabicae* Yu Song, K. Geng, K.D. Hyde and Yong Wang bis, in Song, Geng, Zhang, Hyde, Zhao, Wei, Kang and Wang, Phytotaxa 126(1): 26 (2013)

Notes: This species is clearly a member of *Neopestalotiopsis* according to the results of phylogenetic analyses (Figure 4). Although it was listed as *N. coffeae-arabicae* in various phylogenies [30,35,41,53,63,68], this combination is neither present in the Index of Fungi nor Mycobank, and could also not be traced in the literature, indicating that it has not been validly published, which is therefore performed here.

***Neopestalotiopsis siciliana* Voglmayr, Fiorenza and Aiello, sp. nov**.—MycoBank MB 844082; (Figure 9).

**Etymology**. Named after the region where it was found (Sicily).

**Holotype**. ITALY, Sicily, Catania Province, Giarre, on stems of *Persea americana*, 15 December 2020, Alberto Fiorenza (WU-MYC 0045982; culture AC46 = CBS 149117).

**Description**—Sexual morph unknown. Asexual morph: Conidiomata on natural substrate acervular, in culture on PDA sporodochial, solitary, pulvinate, black, and (100–)300–2000(–2800) μm diam., exuding black, globose, and glistening conidial masses. Conidiophores indistinct, usually reduced to conidiogenous cells. Conidiogenous cells 7.7–15.2 × 2.8–6.7 μm, discrete, cylindrical, ampulliform to lageniform, hyaline, smooth, thin-walled, simple, holoblastic-annelidic, and proliferating one to two times percurrently, with collarette present and not flared. Conidia (20–)23–27(–32) × (6–)6.8–7.9(–8.8) μm, l/w = (2.8–)3.1–3.8(–4.9) (*n* = 102), fusoid, straight or slightly curved, four-septate, smooth, and slightly constricted at the septa; the basal cell obconic with a truncate base, thin-walled, hyaline or pale brown, (3–)4.3–6(–7.2) μm long; three median cells trapezoid or subcylindrical, (12–)14–17(–23) μm long, smooth-walled, versicolored, with septa darker than the rest of cell; the second cell from the base pale brown and (3.8–)4.5–5.4(–6.4) μm long; the third cell medium brown and (4.1–)4.5–5.5(–7) μm long; the fourth cell medium brown and (3.9–)4.6–5.7(–6.5) μm long; with septum between the third and fourth cell more distinct, broader, and darker brown than the other septa; the apical cell conic with a subacute apex, thin-walled, smooth, hyaline, (3.1–)4.1–5.3(–7) μm long, and with two to four apical appendages (mostly three) arising from the apical crest; apical appendages unbranched, tubular, centric, and straight or slightly bent, (19–)24–34(–38) μm long and (0.9–)1.1–1.5(–1.7) μm wide (*n* = 105); basal appendage single, filiform, unbranched, centric, (2.8–)4.6–9.3(–15.3) μm long, and (0.5–)0.7–0.9(–1.1) μm wide (*n* = 85). 

**Culture characteristics**. Colony on PDA attaining 90 mm diameter after 7 days at 24.6 °C, dirty white, with fluffy white aerial mycelium, conidiomata scattered, isolated (Figure 8C). Reverse pale buff (Figure 8D).

**Habitat**. On stems of *Persea americana* Mill. 

**Distribution**. Sicily, Italy.

**Specimens examined**. ITALY, Sicily, Catania Province, Giarre, 15 December 2020, collector Alberto Fiorenza (WU-MYC 0045983; culture AC48 = CBS CBS 149118); Giarre, 15 December 2020, collector Alberto Fiorenza (culture AC49 = CBS 149119).

**Notes**. The phylogenetic analyses revealed a highly supported, distinct phylogenetic position within *Neopestalotiopsis*, confirming its status as a new species. Remarkably, it clusters with four unnamed *Neopestalotiopsis* accessions isolated from the leaves and fruits of *Eriobotrya japonica* in Japan [55]. While one of these strains has sequences identical to our isolates, with which it clustered within a highly supported subclade, the three other strains from *Eriobotrya* form a sister clade to the former (Figure 4). As there are some molecular differences between these two subclades, it is yet unclear whether one or two species are involved; considering the uncertainties in species circumscription in the genus and the lack of morphological data, we here maintain these isolates as *Neopestalotiopsis* sp. As usually observed in *Neopestalotiopsis*, it is impossible to identify *N. siciliana* by conidial morphology alone, and sequence data are necessary for reliable species identification.

## 5. Discussion

The results of this study confirm the presence of *Neopestalotiopsis* species causing disease on young avocado plants in southern Italy. The fungal species obtained from symptomatic tissues were identified based on the morphological characteristics and molecular phylogenetic analyses of the ITS, *TEF1*, and *TUB2* gene regions. The phylogenetic analyses showed that two species are involved in avocado stem and wood lesions, resulting in the dieback of the plants. Of the four isolates sequenced, one was identified as *N. rosae*, while another three isolates formed a clade distinct from the other described *Neopestalotiopsis* species, which is therefore here described as a new species, *N. siciliana*. Remarkably, these three isolates were phylogenetically close to four unnamed *Neopestalotiopsis* strains isolated from *Eriobotrya japonica* in Japan [55]; one even had identical sequences to our isolates. This demonstrates that *N. siciliana* is widespread and has a wider host range. There are a few reports from avocados attributable to the genus *Neopestalotiopsis*, for which ITS sequences are available. Valencia et al. [20] recorded *N. clavispora* (as *Pestalotiopsis clavispora*; ITS sequence HQ659767) as a causal agent of post-harvest stem-end rot in Chile, while Shetty et al. [69] identified one of their endophytic isolates from organically grown avocado trees in South Florida as *Neopestalotiopsis foedans* (ITS sequence KU593530). A sequence comparison of the ITS sequences with our matrix showed that the ITS sequence HQ659767 was identical and that KU593530 was almost identical (one gap difference) to our isolate AC50 representing *N. rosae*. While this could indicate that *N. rosae* is regularly found on avocado, it needs to be noted that the ITS alone is not suitable for species identification, as several species (e.g., *N. hispanica*, *N. longiappendiculata*, *N. mesopotamica*, *N. scalabiensis*, and *N. vaccinii*) have ITS sequences identical to those of *N. rosae*. Therefore, the species identity of these avocado isolates remains uncertain in the lack of *TEF1* and *TUB2* sequences.

*Pestalotiopsis sensu lato* was recently revised by Maharachchikumbura et al. [21] and segregated into three distinct genera, viz. *Pestalotiopsis*, *Pseudopestalotiopsis*, and *Neopestalotiopsis*. During our field surveys, we often encountered young plants of avocado showing stunted growth. Monitoring the plants after the transplant from the nurseries to the open field, it was noticed that some of these were not able to survive. Deeper observations of the internal tissues revealed frequent necrosis and cankers at the grafting point. Likely, propagation processes represent relevant infection courts for pestalotioid fungi. Most of the avocado plants transplanted in Sicily derive from Spanish nurseries where the propagation steps are performed. It is not unusual that symptoms such as stunting, shoot blight, and cankers observed by the growers after the first years from the transplant in the open field are the results of previous infections that had occurred in the nurseries, especially during the grafting. In fact, wounds and injuries are crucial for penetration of the host tissue and subsequent development of the infection, especially for pestalotioid species [70]. Our observations are indeed in accordance with other reports. In China, 30% of symptomatic avocado plants in a nursery plantation showed the presence of *Pestalotiopsis longiseta* [71] and other authors also reported the presence of *Pestalotiopsis* spp. at the graft union in different hosts [72,73,74]. Species of these genera are widely distributed in tropical and temperature areas. This group of fungi is commonly found as endophytes and plant pathogens on different hosts, causing stem-end rot, stem and leaf blight, trunk disease, and cankers [21,75]. Previous investigation conducted in Sicily already reported the presence of pestalotioid fungi, including *Pestalotiopsis clavispora* and *P. uvicola*, causing diseases on tropical, as well as on ornamental, hosts [22,75]. This study represents a new step forward in the insight of this complex and still understudied group of fungi, especially within the genus of *Neopestalotiopsis*, providing new information on the ecology of these two species.

## 6. Conclusions

Two fungal species, *Neopestalotiopsis rosae* and *N. siciliana* sp. nov. are described and illustrated. These fungi were isolated from the stem tissues of diseased young avocado plants in Sicily, Italy. Pathogenicity tests were performed, and Koch’s postulates were fulfilled. The result of this study provided new information regarding this still understudied group of phytopathogenic fungi and its wide host range. *Neopestalotiopsis rosae* and *N. siciliana* could be a new threat to the Italian avocado industry, especially in the southern regions where avocado represents an emerging crop. The presence of these species in the internal tissues at the graft union corroborates the fact that the propagation processes represent crucial steps to obtain healthy material. Further investigations are needed in order to ascertain the diffusion and epidemiology of these species in the Sicilian avocado orchards, and to evaluate the effective risk for the industry. Therefore, it will be important to carry out additional studies on the pathogenicity and susceptibility of the different cultivars of avocado in the future. To our knowledge, this is the first report of *N. rosea* and of the fungus here described as *N. siciliana* affecting avocado.

## Figures and Tables

**Figure 1 jof-08-00562-f001:**
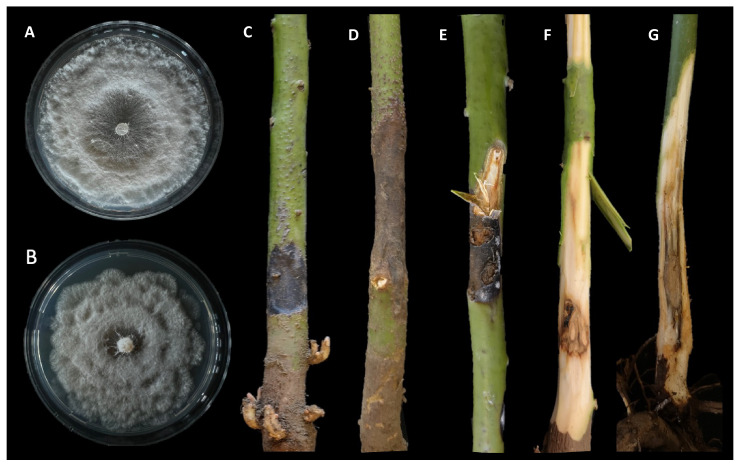
Symptoms caused by *Neopestalotiopsis* spp. on avocado. (**A**) Colony of *N. rosae* isolate AC50 grown on PDA for 7 days; (**B**) colony of *N. siciliana* isolate AC46 grown on PDA for 7 days; (**C**) external lesion; (**D**) shrinkage of the necrotic tissue; (**E**) external lesion with bark cracking; (**F**,**G**) wood discoloration and brownish streaking.

**Figure 2 jof-08-00562-f002:**
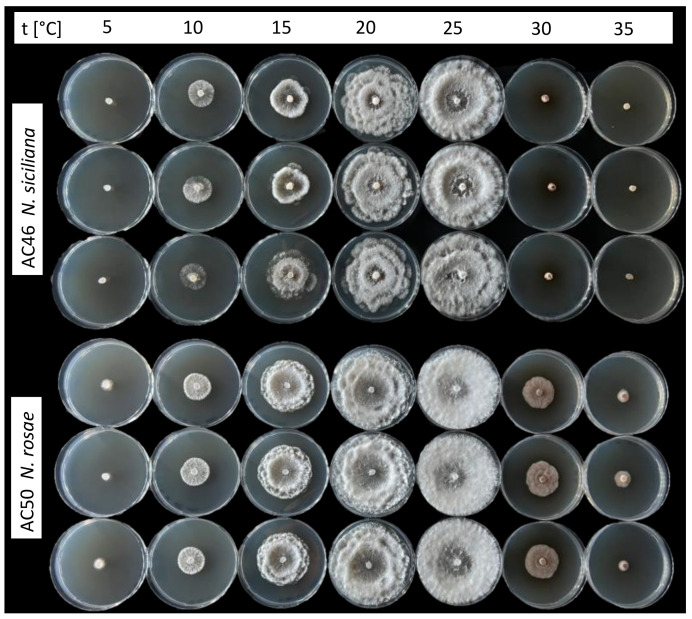
Effect of temperature on mycelial growth rate of two *Neopestalotiopsis* spp. isolated from avocado after 7 days of incubation.

**Figure 3 jof-08-00562-f003:**
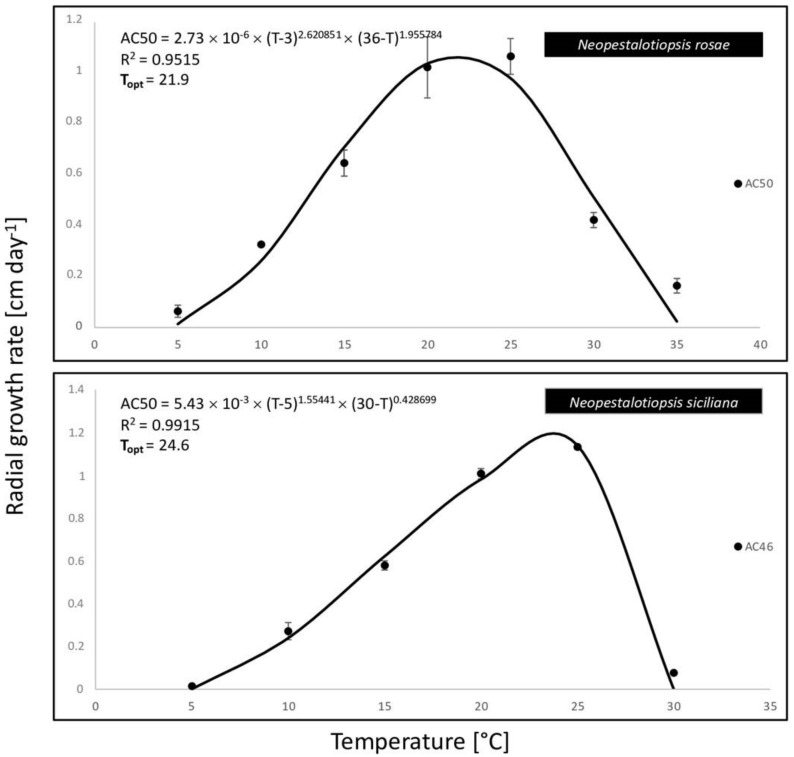
Effect of temperature on mycelial growth rate of two *Neopestalotiopsis* spp. isolated from avocado. The averages of radial growth rate and temperatures were adjusted to a nonlinear regression curve through the Analytis β model. Data points are the means of two independent experiments of three replicated Petri dishes. Vertical bars are the standard error of the means.

**Figure 4 jof-08-00562-f004:**
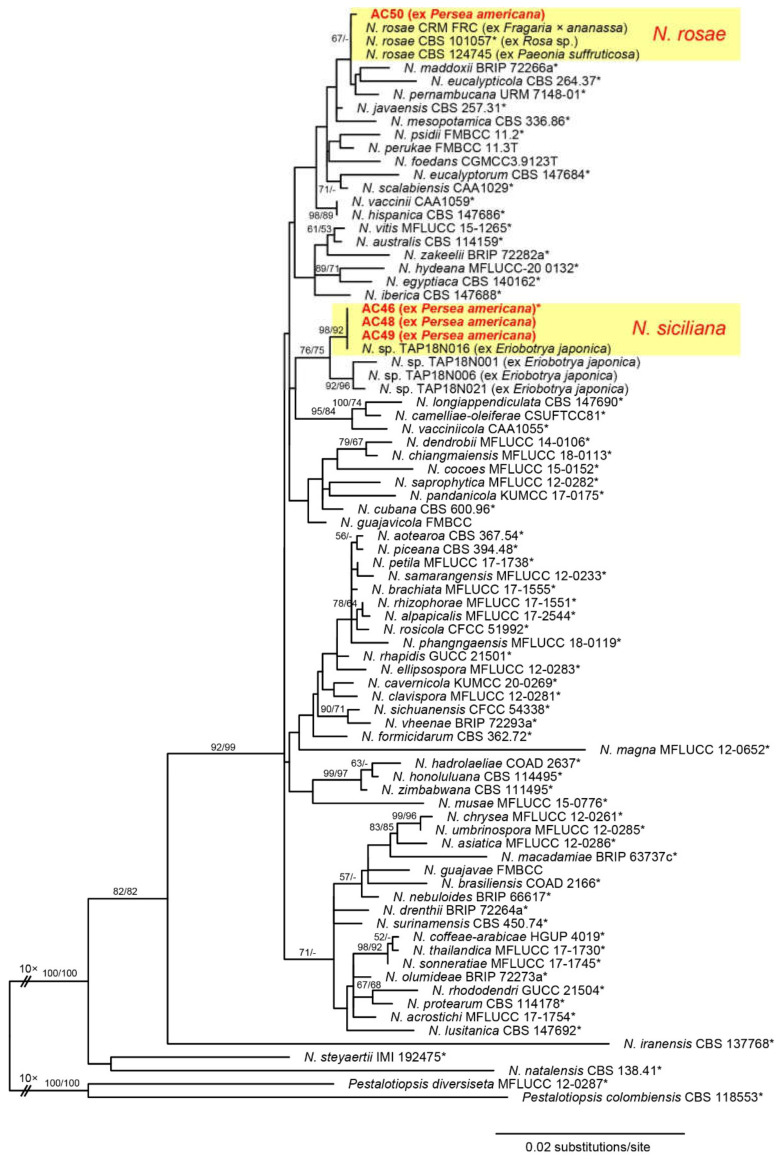
Phylogram of the best ML tree (−lnL = 8699.596) revealed by RAxML from an analysis of the combined ITS-*TEF1*-*TUB2* matrix of *Neopestalotiopsis*, showing the phylogenetic position of the isolates obtained from diseased avocado stem tissue (bold red). Strains marked by an asterisk (*) represent ex-type strains. ML and MP bootstrap support above 50% are given above or below the branches. The broken branches were scaled to one tenth.

**Figure 5 jof-08-00562-f005:**
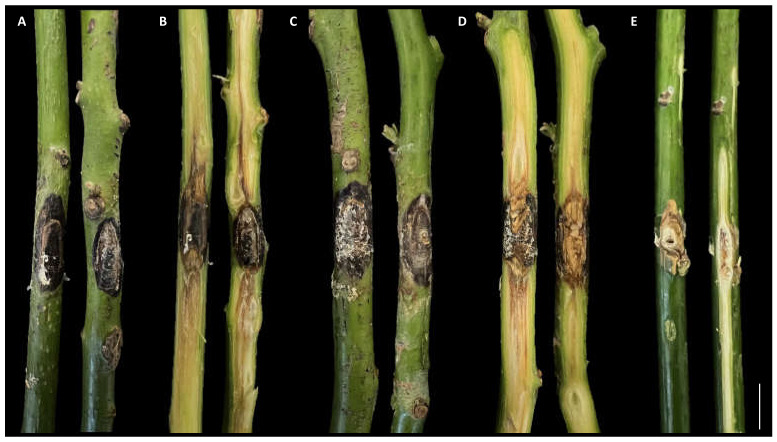
Results of pathogenicity test after 50 days. (**A**,**B**) External and internal lesions caused by *Neopestalotiopsis rosae*; (**C**,**D**) external and internal lesions caused by *N. siciliana*; (**E**) control. Scale bar = 2 cm.

**Figure 6 jof-08-00562-f006:**
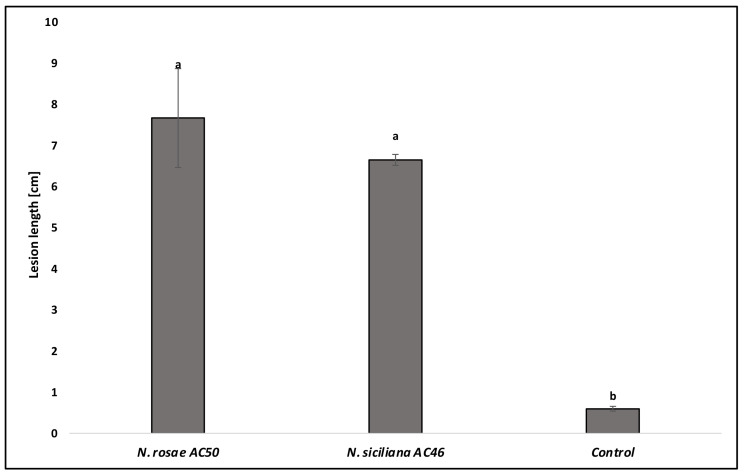
Comparisons of average lesion length (cm) resulting from pathogenicity tests among *Neopestalotiopsis rosae* and *N. siciliana* on potted plants. Columns are the means of 6 inoculation points (2 per plants) for each fungal species. Control consisted of 6 inoculation points. Vertical bars represent the standard error of the means. Bars topped with different letters indicate treatments that were significantly different according to Fisher’s protected LSD test (α = 0.05).

**Figure 7 jof-08-00562-f007:**
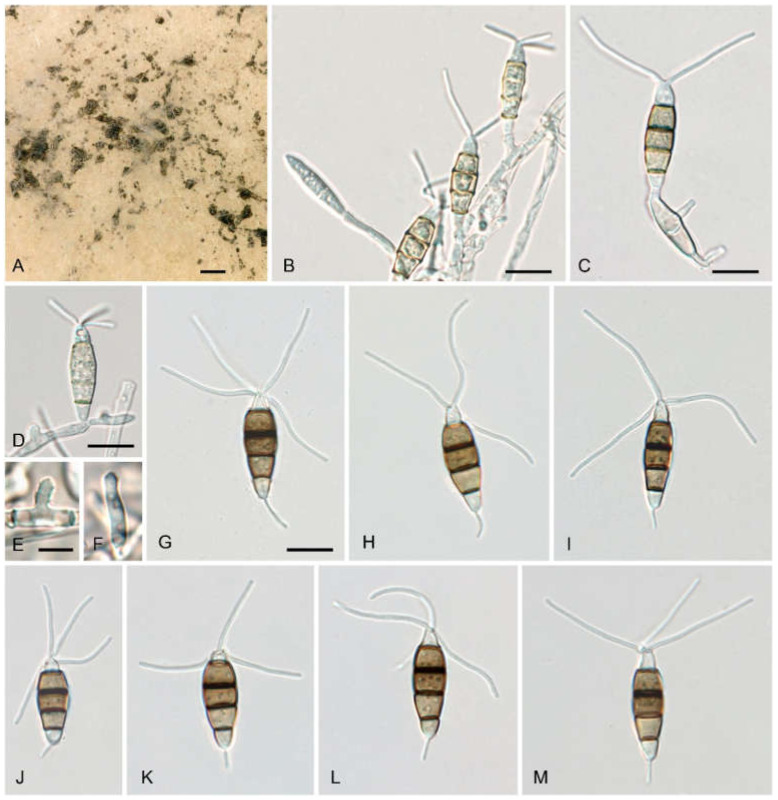
*Neopestalotiopsis rosae* (strain AC50). (**A**) PDA culture with sporodochial conidiomata and black conidial masses; (**B**–**D**) conidiogenous cells giving rise to conidia; (**E**,**F**) holoblastic-annelidic conidiogenous cells; (**G**–**M**) conidia. All in tap water. Scale bars: (**A**) = 200 μm; (**B**–**D**,**G**–**M**) = 10 μm, (**E**,**F**) = 5 μm.

**Figure 8 jof-08-00562-f008:**
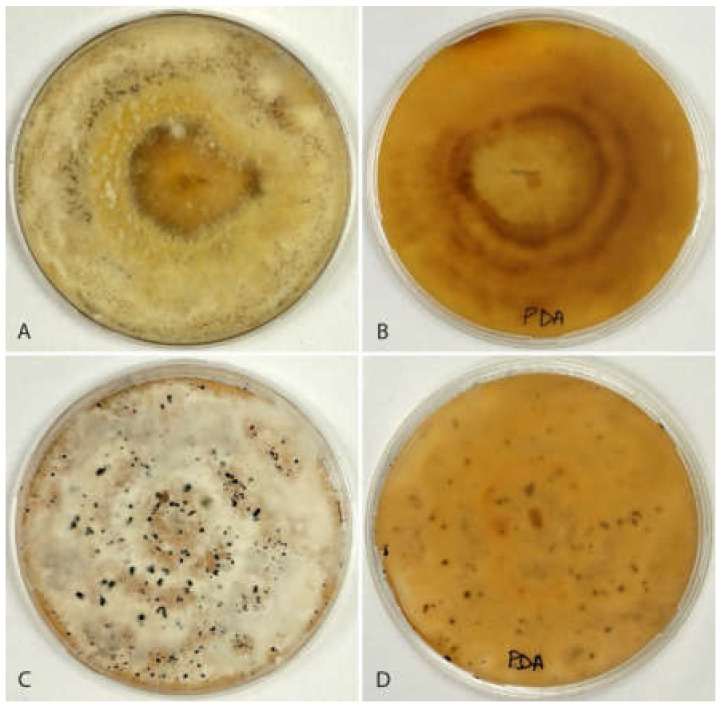
Cultures of *Neopestalotiopsis* spp. from avocado on PDA after 4 weeks. (**A**,**B**) *N. rosae* isolate AC50 from top (**A**) and reverse (**B**); (**C**,**D**) *N. siciliana* isolate AC46 from top (**C**) and reverse (**D**).

**Figure 9 jof-08-00562-f009:**
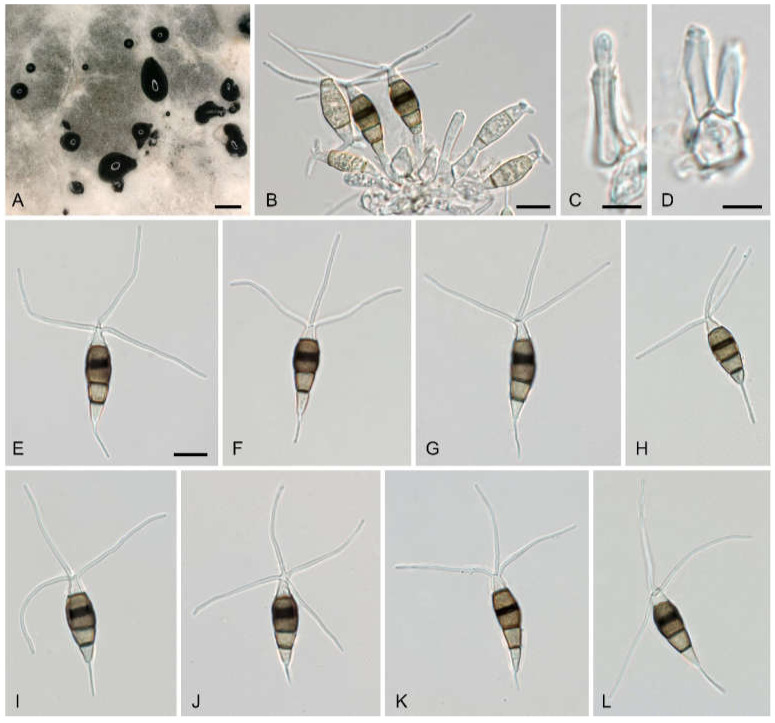
*Neopestalotiopsis siciliana* (**A**–**G**,**I**–**L**) strain AC46, holotype; (**H**) strain AC48). (**A**) PDA culture with sporodochial conidiomata and drops of black conidial masses; (**B**) conidiogenous cells giving rise to conidia; (**C**,**D**) holoblastic-annelidic conidiogenous cells; (**E**–**L**) conidia. All in tap water. Scale bars: (**A**) = 1 mm; (**B**,**E**–**L**) = 10 μm, (**C**,**D**) = 5 μm.

**Table 1 jof-08-00562-t001:** Information of fungal isolates used in the phylogenetic analysis and their corresponding GenBank accession numbers. Isolates in bold are from this study.

Species	Strain ^1^	Host/Substrate	Origin	GenBank Accession Numbers ^2^			Reference
				ITS	*TEF1*	*TUB2*	
*Neopestalotiopsis acrostichi*	MFLUCC 17-1754 ^T^	*Acrostichum aureum*	Thailand	MK764272	MK764316	MK764338	[30]
*N. alpapicalis*	MFLUCC 17-2544 ^T^	*Rhizophora mucronata*	Thailand	MK357772	MK463547	MK463545	[31]
*N. aotearoa*	CBS 367.54 ^T^	Canvas	New Zealand	KM199369	KM199526	KM199454	[32]
*N. asiatica*	MFLUCC 12-0286 ^T^	Unidentified tree	China	JX398983	JX399049	JX399018	[32]
*N. australis*	CBS 114159 ^T^	*Telopea* sp.	Australia	KM199348	KM199537	KM199348	[21]
*N. brachiata*	MFLUCC 17-1555 ^T^	*Rhizophora apiculata*	Thailand	MK764274	MK764318	MK764340	[30]
*N. brasiliensis*	COAD 2166 ^T^	*Psidium guajava*	Brazil	MG686469	MG692402	MG692400	[33]
*N. camelliae-oleiferae*	CSUFTCC81 ^T^	*Camellia oleifera*	China	OK493585	OK507955	OK562360	[34]
*N. cavernicola*	KUMCC 20-0269 ^T^	Cave rock surface	China	MW545802	MW550735	MW557596	[35]
*N. chiangmaiensis*	MFLUCC 18-0113 ^T^	Dead leaves	Thailand	N/A	MH388404	MH412725	[36]
*N. chrysea*	MFLUCC 12-0261 ^T^	*Pandanus* sp.	China	JX398985	JX399051	JX399020	[32]
*N. clavispora*	MFLUCC 12-0281 ^T^	*Magnolia* sp.	China	JX398979	JX399045	JX399014	[32]
*N. cocoes*	MFLUCC 15-0152 ^T^	*Cocos nucifera*	Thailand	NR_156312	KX789689	N/A	[30]
*N. coffeae-arabicae*	HGUP 4019 ^T^	*Coffea arabica*	China	KF412649	KF412646	KF412643	[37]
*N. cubana*	CBS 600.96 ^T^	Leaf litter	Cuba	KM199347	KM199521	KM199438	[21]
*N. dendrobii*	MFLUCC 14-0106 ^T^	*Dendrobium cariniferum*	Thailand	MK993571	MK975829	MK975835	[38]
*N. drenthii*	BRIP 72264a ^T^	*Macadamia integrifolia*	Australia	MZ303787	MZ344172	MZ312680	[39]
*N. egyptiaca*	CBS 140162 ^T^	*Mangifera indica*	Egypt	KP943747	KP943748	KP943746	[40]
*N. ellipsospora*	MFLUCC 12-0283 ^T^	Dead plant materials	China	JX398980	JX399047	JX399016	[32]
*N. eucalypticola*	CBS 264.37 ^T^	*Eucalyptus globulus*	N/A	KM199376	KM199551	KM199431	[21]
*N. eucalyptorum*	CBS 147684 ^T^	*Eucalyptus globulus*	Portugal	MW794108	MW805397	MW802841	[41]
*N. foedans*	CGMCC 3.9123 ^T^	Mangrove plant	China	JX398987	JX399053	JX399022	[32]
*N. formicidarum*	CBS 362.72 ^T^	Dead Formicidae (ant)	Ghana	KM199358	KM199517	KM199455	[21]
*N. guajavae*	FMBCC 11.1 ^T^	*Psidium guajava*	Pakistan	MF783085	MH460868	MH460871	[42]
*N. guajavicola*	FMBCC 11.4 ^T^	*Psidium guajava*	Pakistan	MH209245	MH460870	MH460873	[42]
*N. hadrolaeliae*	COAD 2637 ^T^	*Hadrolaelia jongheana*	Brazil	MK454709	MK465122	MK465120	[43]
*N. hispanica*	CBS 147686 ^T^	*Eucalyptus globulus*	Portugal	MW794107	MW805399	MW802840	[41]
*N. honoluluana*	CBS 114495 ^T^	*Telopea* sp.	USA	KM199364	KM199548	KM199457	[21]
*N. hydeana*	MFLUCC 20-0132 ^T^	*Artocarpus heterophyllus*	Thailand	MW266069	MW251129	MW251119	[44]
*N. iberica*	CBS 147688 ^T^	*Eucalyptus globulus*	Portugal	MW794111	MW805402	MW802844	[41]
*N. iranensis*	CBS 137768 ^T^	*Fragaria × ananassa*	Iran	KM074048	KM074051	KM074057	[45]
*N. javaensis*	CBS 257.31 ^T^	*Cocos nucifera*	Indonesia	KM199357	KM199548	KM199457	[21]
*N. longiappendiculata*	CBS 147690 ^T^	*Eucalyptus globulus*	Portugal	MW794112	MW805404	MW802845	[41]
*N. lusitanica*	CBS 147692 ^T^	*Eucalyptus globulus*	Portugal	MW794110	MW805406	MW802843	[41]
*N. macadamiae*	BRIP 63737c ^T^	*Macadamia integrifolia*	Australia	KX186604	KX186627	KX186654	[46]
*N. maddoxii*	BRIP 72266a ^T^	*Macadamia integrifolia*	Australia	MZ303782	MZ344167	MZ312675	[39]
*N. magna*	MFLUCC 12-0652 ^T^	*Pteridium* sp.	France	KF582795	KF582791	KF582793	[47]
*N. mesopotamica*	CBS 336.86 ^T^	*Pinus brutia*	Turkey	KM199362	KM199555	KM199441	[21]
*N. musae*	MFLUCC 15-0776 ^T^	*Musa* sp.	Thailand	NR_156311	KX789685	KX789686	[30]
*N. natalensis*	CBS 138.41 ^T^	*Acacia mollissima*	South Africa	NR_156288	KM199552	KM199466	[47]
*N. nebuloides*	BRIP 66617 ^T^	*Sporobolus jacquemontii*	Australia	MK966338	MK977633	MK977632	[48]
*N. olumideae*	BRIP 72273a ^T^	*Macadamia integrifolia*	Australia	MZ303790	MZ344175	MZ312683	[39]
*N. pandanicola*	KUMCC 17-0175 ^T^	*Pandanus* sp.	China	N/A	MH388389	MH412720	[36]
*N. pernambucana*	URM 7148-01 ^T^	*Vismia guianensis*	Brazil	KJ792466	KU306739	N/A	[49]
*N. perukae*	FMBCC 11.3 ^T^	*Psidium guajava*	Pakistan	MH209077	MH523647	MH460876	[42]
*N. petila*	MFLUCC 17-1738 ^T^	*Rhizophora apiculata*	Thailand	MK764276	MK764320	MK764342	[30]
*N. phangngaensis*	MFLUCC 18-0119 ^T^	*Pandanus* sp.	Thailand	MH388354	MH388390	MH412721	[36]
*N. piceana*	CBS 394.48 ^T^	*Picea* sp.	UK	KM199368	KM199527	KM199453	[21]
*N. protearum*	CBS 114178 ^T^	*Leucospermum cuneiforme*	Zimbabwe	JN712498	KM199542	KM199463	[50]
*N. psidii*	FMBCC 11.2 ^T^	*Psidium guajava*	Pakistan	MF783082	MH460874	MH477870	[42]
*N. rhapidis*	GUCC 21501 ^T^	*Rhododendron simsii*	China	MW931620	MW980442	MW980441	[51]
*N. rhizophorae*	MFLUCC 17-1551 ^T^	*Rhizophora mucronata*	Thailand	MK764277	MK764321	MK764343	[30]
*N. rhododendri*	GUCC 21504 ^T^	*Rhododendron simsii*	China	MW979577	MW980444	MW980443	[51]
*N. rosae*	CBS 101057 ^T^	*Rosa* sp.	New Zealand	KM199359	KM199523	KM199429	[21]
*N. rosae*	CBS 124745	*Paeonia suffruticosa*	USA	KM199360	KM199524	KM199430	[21]
*N. rosae*	CRM-FRC	*Fragaria × ananassa*	Mexico	MN385718	MN268532	MN268529	[52]
*N. rosae*	**AC50**	*Persea americana*	**Italy**	**ON117810**	**ON107276**	**ON209165**	**this study**
*N. rosicola*	CFCC 51992 ^T^	*Rosa chinensis*	China	KY885239	KY885243	KY885245	[30]
*N. samarangensis*	MFLUCC 12-0233 ^T^	*Syzygium samarangense*	Thailand	JQ968609	JQ968611	JQ968610	[30]
*N. saprophytica*	MFLUCC 12-0282 ^T^	*Magnolia* sp.	China	JX398982	JX399048	JX399017	[21]
*N. scalabiensis*	CAA1029 ^T^	*Vaccinium corymbosum*	Portugal	MW969748	MW959100	MW934611	[53]
*N. sichuanensis*	CFCC 54338 ^T^	*Castanea mollissima*	China	MW166231	MW199750	MW218524	[54]
*N. siciliana*	**AC46**	*Persea americana*	**Italy**	**ON117813**	**ON107273**	**ON209162**	**this study**
*N. siciliana*	**AC48**	*Persea americana*	**Italy**	**ON117812**	**ON107274**	**ON209163**	**this study**
*N. siciliana*	**AC49**	*Persea americana*	**Italy**	**ON117811**	**ON107275**	**ON209164**	**this study**
*N. sonneratiae*	MFLUCC 17-1745 ^T^	*Sonneronata alba*	Thailand	MK764280	MK764324	MK764346	[30]
*N.* sp.	TAP18N001	*Eriobotrya japonica*	Japan	LC427126	LC427128	LC427127	[55]
*N.* sp.	TAP18N006	*Eriobotrya japonica*	Japan	LC427141	LC427143	LC427142	[55]
*N.* sp.	TAP18N016	*Eriobotrya japonica*	Japan	LC427168	LC427170	LC427169	[55]
*N.* sp.	TAP18N021	*Eriobotrya japonica*	Japan	LC427183	LC427185	LC427184	[55]
*N. steyaertii*	IMI 192475 ^T^	*Eucalytpus viminalis*	Australia	KF582796	KF582792	KF582794	[47]
*N. surinamensis*	CBS 450.74 ^T^	Soil under *Elaeis guineensis*	Suriname	KM199351	KM199518	KM199465	[21]
*N. thailandica*	MFLUCC 17-1730 ^T^	*Rhizophora mucronata*	Thailand	MK764281	MK764325	MK764347	[30]
*N. umbrinospora*	MFLUCC 12-0285 ^T^	unidentified plant	China	JX398984	JX399050	JX399019	[32]
*N. vaccinii*	CAA1059 ^T^	*Vaccinium corymbosum*	Portugal	MW969747	MW959099	MW934610	[53]
*N. vacciniicola*	CAA1055 ^T^	*Vaccinium corymbosum*	Portugal	MW969751	MW959103	MW934614	[53]
*N. vheenae*	BRIP 72293a ^T^	*Macadamia integrifolia*	Australia	MZ303792	MZ344177	MZ312685	[39]
*N. vitis*	MFLUCC 15-1265 ^T^	*Vitis vinifera*	China	KU140694	KU140676	KU140685	[56]
*N. zakeelii*	BRIP 72282a ^T^	*Macadamia integrifolia*	Australia	MZ303789	MZ344174	MZ312682	[39]
*N. zimbabwana*	CBS 111495 ^T^	*Leucospermum cuneiforme*	Zimbabwe	MH554855	KM199545	KM199456	[21]
*Pestalotiopsis colombiensis*	CBS 118553 ^T^	*Eucalyptus grandis* × *urophylla*	Colombia	KM199307	KM199488	KM199421	[21]
*Pestalotiopsis diversiseta*	MFLUCC 12-0287 ^T^	Dead plant material	China	NR_120187	JX399073	JX399040	[32]

^1^ BRIP: Queensland Plant Pathology Herbarium, Australia; CAA: Personal culture collection of Artur Alves, Department of Biology, University of Aveiro; CBS: Culture collection of the Westerdijk Fungal Biodiversity Institute, Utrecht, The Netherlands; CFCC: China Forestry Culture Collection Center, Research Institute of Forest Ecology, Environment and Protection, Beijing, China; CGMCC: China General Microbiological Culture Collection Center, Institute of Microbiology, Chinese Academy of Sciences, Beijing, China; COAD: Culture collection of Coleção Octávio Almeida Drummond of the Universidade Federal de Viçosa, Viçosa, Brazil; CRM: Universidad Autónoma Chapingo, Centro Regional Morelia, Morelia, Michoacán, México; CSUFTCC: Central South University of Forestry and Technology culture collection, Hunan, China; FMBCC: Fungal Molecular Biology Laboratory Culture Collection, University of Agriculture Faisalabad, Pakistan; GUCC: Department of Plant Pathology culture collection, Agriculture College, Guizhou University, China; HGUP: Plant Pathology Herbarium of Guizhou University, Guizhou, China; IMI: Culture collection of CABI Europe UK Centre, Egham, UK; KUMCC: Culture collection of Kunming Institute of Botany, Chinese Academy of Sciences, Kunming, China; MFLUCC: Mae Fah Luang University culture collection, Chiang Rai, Thailand; TAP: Culture collection of Tamagawa University, Tokyo, Japan; URM: The Father Camille Torrend Herbarium, Pernambuco, Brazil. Ex-type strains are labeled with ^T^. ^2^ ITS: internal transcribed spacer; *TEF1*: translation elongation factor 1-α; *TUB2*: β-tubulin. N/A: Not available.

## Data Availability

The data presented in this study are available on request from the corresponding author.

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
