# Peer review of "Neopestalotiopsis siciliana sp. nov. and N. rosae Causing Stem Lesion and Dieback on Avocado Plants in Italy"

_jof, 2022, doi:10.3390/jof8060562_

Round 1
Reviewer 1 Report
Dear Authors,
The current study seems very interesting but I would like to ask whether you obtain cultures of any other common fungal species (such as Alternaria sp. etc) while you are isolating the Neopestalotiopsis.
Very few comments were made in the manuscript and I wish best for the authors.
Thank you.

Author Response
Thanks for your comments and suggestions. We revised the manuscript as suggested. The answers to comments are listed below:
Q The current study seems very interesting but I would like to ask whether you obtain cultures of any other common fungal species (such as Alternaria sp. etc) while you are isolating the Neopestalotiopsis.
A We obtained 90% of Neopestalotiopsis from infected tissues, other occasional fungi were not considered.
Q Please mention how many samples/isolates did you obtain?
A About one hundred small pieces of the stem from 20 plants were used for isolation. Different isolates of Neopestalotiopsis were obtained from infected tissues but only 8 single spore isolates of Neopestalotiopsis were collected and kept in our fungal collection. Please see the M&M and Results of Isolation and Morphological Characterization.
Reviewer 2 Report
This study shows appropriate evidence that Dieback Disease in Avocado is caused by two species of the genus Neopetalotiopsis. The research design is followed by Koch’s postulate. Authors identified pathogens which were N. rosae and new species. However, I found several inappropriate points as described below.
Major points:
1) Your N. rosae isn’t aggregated as one clade with other N. rosae strains. Even though any N. rosae strains don’t fall into one clade with an ex-type strain in the phylogenic tree, this just indicates that the previous study may be not correct. Authors shouldn’t follow the previous wrong result. Thus, I don’t agree with author’s logic to identify AC50 strain as N. rosae.
As part of this matter, I hope you change your speculation in Discussion.
2) Regarding new species, author is required to describe the difference to new species morphologically from the existing species of the genus Neopestalotiopsis in the section of “Note”. For comparison, â‘ those of existing species with similar size of conidia with your type specimen, â‘¡species belonging to sister clades, and â‘¢species isolated from avocado are selected usually. When the size of your strains was overlapped with other species, your strains would be treated as “cryptic species”, not be “new species”.
However, some researchers in current study show “new (phylogenetic) species” instead of “cryptic species”, describe the reason to establish new species, and mention the sequences data as evidence to establish a new species in “Note”.
3) Identification and Taxonomy differ and have each definition. Taxonomy part should be written about new species and new combine information. Others are written as identification.
(Thus, L 239- 279 move to other part)
4) I recommend you add information if both species produce same symptoms to a tree of avocado.
Other minor points:
L196- 200
Neopestalotiopsis, N. rosae, and Eriobotrya japonica should be italicized.
Author Response
Thanks for your suggestions. We revised the manuscript. The answers to comments are listed below:
Q Your N. rosae isn’t aggregated as one clade with other N. rosae strains. Even though any N. rosae strains don’t fall into one clade with an ex-type strain in the phylogenic tree, this just indicates that the previous study may be not correct. Authors shouldn’t follow the previous wrong result. Thus, I don’t agree with author’s logic to identify AC50 strain as N. rosae.
As part of this matter, I hope you change your speculation in Discussion.
A This statement is incorrect and probably based on erroneous interpretations of a phylogram. Three isolates of N. rosae have been selected from Genbank for inclusion in the phylogenetic analyses, one of which is the ex-holotype strain (CBS 101057). ALL of these three strains (CBS 101057, CRM FRC, CBS 124745 ) have COMPLETELY IDENTICAL ITS, tef1 and beta tubulin sequences, so, if one accepts these three markers as reliable for species distinction in Neopestalotiopsis, which is the case in all recently published taxonomic articles on the genus, there is no doubt that these isolates are conspecific. We are aware that some of the N. rosae isolates are misidentified in GenBank, but for this reason none of these has been included in our analysis. Our isolate has completely identical ITS and tef1 sequences to the ex-type strain of N. rosae, and a single bp difference in beta tubulin. There is absolutely no reason to assume that this single bp difference is sufficient for establishment of a new species.
That the N. rosae isolates do not form a clade in the phylogram is caused by the tree-building algorithm how branch lengths are calculated. In fact, such effects are commonly seen in phylogenetic analyses, depending rather on the sequences of the sister clade causing a branch length of zero, due to which accessions with completely identical sequences apparently do not form a clade. By the way, these four N. rosae isolates (including ours) included in our analyses do form a clade in the cladogram (i.e. not showing branch lengths), which is even supported by 68% bootstrap support; however, as the branch length is zero, it is not apparent in the phylogram.
Concerning the discussion, there is no need of any changes, as these “speculations” just make the point that the true identity of the other isolates reported from avocado for which only ITS sequences are available currently cannot be evaluated, simply due to the fact that the ITS alone is not providing enough resolution for species identification.
Q Regarding new species, author is required to describe the difference to new species morphologically from the existing species of the genus Neopestalotiopsis in the section of “Note”. For comparison, those of existing species with similar size of conidia with your type specimen, species belonging to sister clades, and species isolated from avocado are selected usually. When the size of your strains was overlapped with other species, your strains would be treated as “cryptic species”, not be “new species”.
However, some researchers in current study show “new (phylogenetic) species” instead of “cryptic species”, describe the reason to establish new species, and mention the sequences data as evidence to establish a new species in “Note”.
A We strongly disagree with this comment. Everybody dealing with pestalotioid fungi knows that it is impossible to reliably identify species by morphology, simply because the majority of these are cryptic species, with a largely overlapping morphology. In addition, the measurements published in most species descriptions are based on single isolates, not accounting for the morphological variability, and the morphological differences commonly claimed in the Notes are often very questionable as they are often very minor, and it is also very commonly seen that species are only compared to those for which one can find some differences, but those with fully matching morphology are simply ignored. In fact, new species in Neopestalotiopsis are always primarily established based on morphology, and then one wants to find some morphological differences – however, these alone would never be enough for establishing new species. To bring it to the point – the main criterion for species identification in this lineage is always molecular data, and no-one uses conidial morphology knowing that this is unreliable due to insufficient consistent morphological differences. In addition, with about 70 currently described species, it becomes ridiculous to compare the new species with all of them, as one will then see a complete morphological continuum – and a genus consisting primarily of cryptic species.
In lack of practical usability for species distinction, we do not see much point in adding morphological comparison with few or even all 70 described Neopestalotiopsis species.
Q Identification and Taxonomy differ and have each definition. Taxonomy part should be written about new species and new combine information. Others are written as identification.
(Thus, L 239- 279 move to other part)
A Agreed, we moved the description to the new results subheading 3.4. Morphological description of Neopestalotiopsis rosae isolates from avocado.
Q I recommend you add information if both species produce same symptoms to a tree of avocado.
A Both species of Neopestalotiopsis identified in this study were pathogenic to avocado and produced similar stem lesions and severe external and internal wood discoloration. Moreover, the mean lesion lengths of both fungal species were significantly different from the control, but not significantly different between them. See the results of pathogenicity test.
Reviewer 3 Report
Dear editors and authors,
Here is the review of the paper titled "Neopestalotiopsis siciliana sp. nov. and N. rosae Causing Stem Lesion, and Dieback on Avocado Plants" written by Alberto Fiorenza & co-authors. The paper aims to provide taxonomic descriptions of two species of Neopestalotiopsis causing Avocado (Persea americana) dieback. One of those species is described here as a new to science (Neopestalotiopsis siciliana). Moreover, Pestalotiopsis coffeae-arabicae was recombined into the genus Neopestalotiopsis according to the results of phylogenetic analysis based on the combined ITS-TEF1-TUB2 gene sequences matrix. The authors used methods of integrative taxonomy including the examination of morphological characters, observations on the effect of different temperatures on mycelial growth rate, and phylogenetic study of molecular characters (ITS-TEF1-TUB2 gene dataset). In addition, pathogenicity tests were performed on young potted avocado trees with N. siciliana and N. rosae and both species were confirmed as pathogenic to avocado.
The methods used and analyses performed are appropriate for this kind of study. Morphological description, phylogenetic study and discussion are exhaustive and cover all needed parts. The English language is very well. The authors followed the newest version of International code of nomenclature for algae, fungi, and plants. Material and Methods section does not contain a description of methods of microscopical examination of cultures. Please include it in the text.
There are a few other remarks/suggestions on the manuscript included in the pdf document attached. Other than that, the paper could be accepted for publication in JoF.
Best, Reviewer

Author Response
Thanks for your comments. We revised the manuscript as suggested.
Answer to comment:
Q methods of microscopical examination of cultures are not described at all. Please include in M&M 'Isolation and Morphological Characterization'.
A: The section on morphology is added to the materials and methods section.